# Compositional Alteration of Gut Microbiota in Psoriasis Treated with IL-23 and IL-17 Inhibitors

**DOI:** 10.3390/ijms24054568

**Published:** 2023-02-26

**Authors:** Yu-Huei Huang, Lun-Ching Chang, Ya-Ching Chang, Wen-Hung Chung, Shun-Fa Yang, Shih-Chi Su

**Affiliations:** 1Institute of Medicine, Chung Shan Medical University, Taichung 402, Taiwan; 2Department of Dermatology, Chang Gung Memorial Hospital, Linkou Branch, Taoyuan 333, Taiwan; 3School of Medicine, Chang Gung University, Taoyuan 333, Taiwan; 4Department of Mathematical Sciences, Florida Atlantic University, Boca Raton, FL 33431, USA; 5Whole-Genome Research Core Laboratory of Human Diseases, Chang Gung Memorial Hospital, Keelung 204, Taiwan; 6Department of Medical Research, Chung Shan Medical University Hospital, Taichung 402, Taiwan

**Keywords:** psoriasis, interleukin-23 inhibitor, guselkumab, interleukin-17 inhibitor, secukinumab, ixekizumab, gut microbiota, metabolic pathway

## Abstract

Alterations in the gut microbiota composition and their associated metabolic dysfunction exist in psoriasis. However, the impact of biologics on shaping gut microbiota is not well known. This study aimed to determine the association of gut microorganisms and microbiome-encoded metabolic pathways with the treatment in patients with psoriasis. A total of 48 patients with psoriasis, including 30 cases who received an IL-23 inhibitor (guselkumab) and 18 cases who received an IL-17 inhibitor (secukinumab or ixekizumab) were recruited. Longitudinal profiles of the gut microbiome were conducted by using 16S rRNA gene sequencing. The gut microbial compositions dynamically changed in psoriatic patients during a 24-week treatment. The relative abundance of individual taxa altered differently between patients receiving the IL-23 inhibitor and those receiving the IL-17 inhibitor. Functional prediction of the gut microbiome revealed microbial genes related to metabolism involving the biosynthesis of antibiotics and amino acids were differentially enriched between responders and non-responders receiving IL-17 inhibitors, as the abundance of the taurine and hypotaurine pathway was found to be augmented in responders treated with the IL-23 inhibitor. Our analyses showed a longitudinal shift in the gut microbiota in psoriatic patients after treatment. These taxonomic signatures and functional alterations of the gut microbiome could serve as potential biomarkers for the response to biologics treatment in psoriasis.

## 1. Introduction

Psoriasis is an inflammatory skin disease that is associated with many other medical conditions, and affects adults and children worldwide [1]. Overall prevalence ranges from 0.1% in east Asia to 1.5% in western Europe, and is highest in high-income countries [2,3]. Most patients with psoriasis have some detriment to their quality of life attributable to the disease, and many feel a substantial, negative effect on their psychosocial wellbeing. It has been regarded that psoriasis involves the interplay between predisposing genetic and environmental (e.g., infection and antibiotics treatment) factors [1,4,5,6]. Studies have shown that skin and the gut microbiome play a role in modulating the development of chronic plaque psoriasis [7]. Recent evidence revealed a combined increase in *Corynebacterium*, *Propionibacterium*, *Staphylococcus*, and *Streptococcus* in psoriatic plaque sites [7,8]. Gut microbiota is known to play a critical role in the regulation of metabolism, the immune system, and intestine permeability [9]. A disturbed intestinal microbiome was shown to be involved in a number of autoimmune diseases including type 1 diabetes, rheumatoid arthritis, multiple sclerosis, celiac disease, and inflammatory bowel disease (IBD) [10,11]. In psoriasis, similar evidence demonstrated gut dysbiosis with lower diversity and altered relative abundance for certain bacteria [9,12]. Several studies have found the relative abundance of *Bacteroidetes* was lower and that of *Firmicutes* was higher in patients with psoriasis compared to healthy controls [12,13,14]. However, an inconsistent result reported by Huang et al., revealed an increased abundance of *Bacteroidetes* and decreased *Firmicutes* in psoriasis [15]. These changes in gut microbiota are considered to be crucial causes for initiating or exacerbating psoriasis in humans and animal models [16,17].

Treatment for psoriasis may change the composition of the skin and gut microbiota [18,19,20]. A change in lesional skin microbiota has been associated with a clinical response after balneotherapy [18] and phototherapy [19]. A reduced mean rate of *Staphylococcus aureus* on psoriatic plaques, reaching a nadir at weeks 16–20 after treatment, was noted in our previous research [20]. Regarding the gut microbial change after psoriasis treatment, the relative abundance of *Pseudomonadaceae* and *Enterobacteriaceae* increased significantly following secukinumab therapy, while no significant change was noted in gut microbiome composition following ustekinumab treatment [21].

In the past 20 years, findings from immunological and genetic studies have highlighted causal immunological circuits of psoriasis that converge on adaptive immune pathways involving interleukin (IL)-17 and IL-23 [1,22,23]. The suppression of psoriasis-related, proinflammatory, and Th17-associated cytokines, such as tumor necrosis factor (TNF)-α, IL-17A, and IL-23, was observed in mice fed with *Lactobacillus pentosus* [24]. The clinical significance of the interaction between microbiota and the immune system is of importance. Although guselkumab, a selective IL-23 inhibitor, and secukinumab and ixekizumab, monoclonal antibodies targeting IL-17A, were highly effective in treating psoriasis, their treatment results in IBD were not consistent. Clinical trials for biologics blocking either IL-17A or its receptor have contributed to the exacerbation of IBD [25,26]. This raised the possibility that blockade of IL-17 could interfere with the microbiota composition and homeostasis in the intestine that might predispose susceptible individuals to develop IBD [27,28]. Moreover, in a phase 2 trial, guselkumab demonstrated a greater efficacy than a placebo in patients with Crohn’s disease [29]. These findings indicated a sophisticated interaction between gut microbiota composition and biologic therapies. Yet, how gut microbiota in psoriasis react to the IL-17 and IL-23 blockers has scarcely been investigated. Therefore, this study aimed to investigate the dynamic alteration of gut microbiota in psoriasis patients before and after receiving IL-17 and IL-23 antagonists.

## 2. Results

### 2.1. Patient Demographic and Characteristics

A total of one hundred and ninety-two fecal samples were obtained from 48 patients with 30 cases receiving the IL-23 inhibitor (guselkumab) (mean age 45.2 years) and 18 cases receiving IL-17 inhibitors (secukinumab and ixekizumab) (mean age 52.8 years). There was no significant difference in gender, weight, psoriatic arthritis, baseline PASI score, and baseline CRP level between the two groups. Patients treated with an IL-17 inhibitor were older than patients treated with an IL-23 inhibitor (Table 1).

The mean PASI scores decreased at weeks 4, 12, and 24 after either IL-23 or IL-17 inhibitor therapy. All these changes from baseline were significant (Figure 1A). In addition, the CRP level was significantly reduced after 12 weeks and 24 weeks of treatment (Figure 1B). Moreover, we found recruited patients did not change their eating habits during the study.

### 2.2. Gut Microbial Diversity in Psoriasis after the Treatment with Il-23 and IL-17 Inhibitors

We studied the temporal alteration of microbial diversity in patients treated with IL-23 or IL-17 inhibitors. Calculation of the weighted-UniFrac distance matrix (β diversity) displayed a significantly altered distance in microbial community structures among samples from patients receiving an IL-23 or IL-17 inhibitor during 24-week treatment, while no significant difference in the α diversity was observed among the groups (Figure 2A). Moreover, Bray–Curtis distance was used to measure β diversity at week 0 and 24 among the responders (R) and non-responders (NR) (Figure 2B). The results showed that β diversity of gut microbiota in the responders to the IL-23 inhibitor was significantly higher than that in non-responders both at baseline and week 24 (*p* < 0.05), while there was no significant difference in β diversity between responders and non-responders treated with IL-17 inhibitors.

### 2.3. Altered Composition of Gut Microbiota in Psoriatic Patients after Treatment with IL-23 and IL-17 Inhibitors

We then sought the most relevant taxa whose abundance altered after the treatment (week 24) to explore the effect of biologics on the composition of gut microbiota. In patients treated with the IL-23 inhibitor, we identified five taxa whose levels were significantly different from the baseline (Figure 3). The relative abundance of *Roseburia*, *Lachnoclostridium*, *Bacteroides vulgatus*, *Anaerostipes*, and *Escherichia–Shigella* increased over the time course of the treatment. In patients treated with IL-17 inhibitors, levels of *Bacteroides stercoris* and *Parabacteroides merdae* were significantly increased at week 24, while those of *Blautia* and *Roseburia* were significantly reduced (Figure 3).

### 2.4. Changes in Relative Abundance of Gut Bacteria between Responders and Non-Responders

Furthermore, we assessed the association between the therapeutic outcome and changes in relative abundance of individual taxa from the baseline to 24 weeks post-treatment. We found that among patients treated with the IL-23 inhibitor for 24 weeks, the relative abundance of *Lachnospiraceae* and *Romboutsia* significantly decreased from the baseline in the responders compared to non-responders (Figure 4). Meanwhile, the relative abundance of *Fusicatenibacter* in patients treated with IL-17 inhibitors for 24 weeks significantly increased compared to non-responders, whereas that of *Lachnospiraceae NK4A136* and *Roseburia* significantly decreased (Figure 4).

### 2.5. Functional Prediction of Gut Microbiome after Treatment with IL-23 and IL-17 Inhibitors

Considering the pathways related to metabolism, we found a number of pathway modules associated with lipid metabolism, inositol phosphate metabolism, and glutathione metabolism enriched in patients treated with the IL-23 inhibitor at week 24. In contrast, bacterial genes assigned to energy metabolism, arginine biosynthesis, cysteine and methionine metabolism, fructose and mannose metabolism, and carbapenem biosynthesis were less abundant. In patients treated with IL-17 inhibitors, the abundance of pathway modules associated with indole alkaloid biosynthesis increased, while that with lysine biosynthesis decreased (Table 2).

In addition, we investigated alterations in microbial functions at week 24 from the baseline between responders and non-responders. Among patients treated with the IL-23 inhibitor, the pathway of taurine and hypotaurine metabolism was enriched in the responders compared to non-responders (Table 3). Among patients treated with IL-17 inhibitors, 13 metabolism pathways were significantly enriched and 3 decreased in responders after 24 weeks of treatment compared with non-responders (Table 3). The pathways involved in amino acids metabolism, biosynthesis of antibiotics, and carbohydrate metabolism were differentially enriched from the baseline after the treatment with IL-17 inhibitors.

## 3. Discussion

In the present study, we analyzed the gut microbial diversities and taxonomies in patients with psoriasis at weeks 4, 12, and 24 after the treatment with IL-23 or IL-17 inhibitors. This is the first study to demonstrate a significant increase in β diversity of gut microbial communities and altered abundance of certain bacteria in patients receiving the IL-23 inhibitor for 24 weeks. In addition, we identified microbial taxa and functional pathways associated with the therapeutic options and treatment responses.

Changes in gut microbiota composition due to therapeutic agents and their influence on clinical response have been reported in patients with inflammatory bowel disease (IBD) [30]. Common types of gut microbiota change after biologics treatment encompassed an increased abundance of short-chain fatty acids (SCFAs)-producing bacteria, which are considered beneficial commensal bacteria [30]. An improvement in intestinal dysbiosis was reported with an increment in the abundance of SCFAs-producing bacteria such as *Anaerostipes*, *Blautia*, and *Roseburia* from IBD patients after receiving infliximab [31]. Moreover, similar findings were also demonstrated in IBD patients receiving ustekinumab [32]. In this study, we found that the relative abundance of *Anaerostipes* and *Roseburia* increased in patients after IL-23 inhibitor treatment, which may increase the production of SCFAs and consequently restore the immunomodulatory function and intestinal epithelial barrier [33,34]. Conversely, the abundance of *Blautia* and *Roseburia* was reduced in those receiving IL-17 inhibitors. One previous study investigating the impact of secukinumab on gut microbial composition [21] showed a reduction in the abundance of the SCFAs-producing bacteria *Firmicutes*, consistent with our findings.

The *Bacteroides* genus constitutes 30% of the total colonic bacteria and *Bacteroides vulgatus* is one of the most commonly encountered *Bacteroides* species in the human gut [35]. The role of *B. vulgatus* in modulating the immune system has been investigated in animal experiments. Supplementation with *B. vulgatus* attenuated symptoms of colitis in mice and decreased the expression of TNF-α, IL-1β, and IL-6 in the colon [36]. Moreover, suppression of the systemic and intestinal immune response was observed in mice gavaged with *Bacteroides vulgatus* [37,38]. The present study demonstrated that the relative abundance of *Bacteroides vulgatus* increased after anti-IL-23 inhibitor treatment, which might further imply the beneficial effect of gut immunomodulation by the IL-23 inhibitor in psoriasis.

The gut is considered to be a major immune organ, with gut-associated lymphoid tissue (GALT) being the most complex immune compartment [39]. It is well known that changes in the gut microbial composition may promote both health and disease [40]. Strong evidence has indicated that intestinal dysbiosis is clinically relevant to psoriasis [41]. The importance of the gut–skin axis in the pathogenesis of psoriasis has recently been documented in humans, as well as in animal models. [42]. In imiquimod-induced psoriasis-like mice, gut microbiota promoted intestinal and cutaneous inflammations by enhancing the IL-23/IL-17 axis [42,43]. In addition, a gut microbial genus, *Romboutsia*, increased in mice with imiquimod-induced psoriasis [43], suggesting that IL-23/IL-17-axis-related psoriasis may be associated with levels of gut *Romboutsia*. Intriguingly, our study revealed that the abundance of *Romboutsia* significantly decreased at week 24 in the responders to the IL-23 inhibitor when compared with non-responders. However, there was no significant difference in the gut *Romboutsia* level between responders and non-responders treated with IL-17 inhibitors. Based on these findings, we speculate that blocking IL-23 may ameliorate *Romboutsia*-mediated psoriasis by improving IL-23/IL-17-axis-related skin inflammation.

At the genus level, an enriched *Lachnospiraceae* NK4A136 group was detected in patients with ankylosing spondylitis [44] and IBD [45]. Recently, a study on the gut microbiome demonstrated an increase in the abundance of gut *Lactobacillaceae* in psoriatic patients [13]. Our results further revealed that the abundance of *Lachnospiraceae NK4A136* at week 24 significantly decreased in responders to IL-17 inhibitors compared to non-responders. It has been shown that the *Lachnospiraceae NK4A136* group is correlated with elevated levels of intestinal IL-17 and IL-6 in mice with diabetes mellitus, resulting in intestinal inflammation [46]. Thus, we hypothesize that responders to IL-17 inhibitors might benefit from reduction in the gut *Lachnospiraceae NK4A136* group, which likely contributes to declined skin inflammation. Further investigation should be conducted to address the causal relationship of these findings.

In our study, sixteen KEGG pathways were found to be significantly enriched in responders to IL-17 inhibitors, such as the biosynthesis of amino acids, energy metabolism, and biosynthesis of antibiotics including vancomycin, validamycin, and novobiocin. Previously, dramatic changes in glucose metabolism, amino acid metabolism, and energy metabolism have been shown in psoriasis [47,48]. Metabolic regulation of cell proliferation and apoptosis was thought to be critical for dysregulated keratinocyte hyperproliferation in psoriasis [49,50]. Altogether, these findings suggest that altered gut-microbiota-mediated biosynthesis of amino acids and energy metabolism may also contribute to specific phenotypes in patients with psoriasis, such as uncontrolled keratinocyte hyperproliferation. It was reported that treatment with broad-spectrum antibiotics in mice with imiquimod-induced psoriasis reduced proinflammatory IL-17-producing T cells and skin thickness [16,42]. Moreover, *Actinobacteria*, isolated from the gut of freshwater fish, exhibited antimicrobial activities by producing antibiotic compounds [51]. Our data showed that gut microbiome-encoded metabolic KEGG pathways enriched in the responders to IL-17 inhibitors were concentrated in the biosynthesis of antibiotics. According to these findings, we suggest that IL-17 inhibitors may partially improve psoriasis-related skin inflammation by enhancing gut-microbiota-mediated biosynthesis of antibiotics.

In addition, reduction in the abundance of the taurine and hypotaurine metabolism pathway in patients with severe psoriasis has been observed in one recent study [52]. Our results demonstrated that the abundance of the taurine and hypotaurine metabolic pathway was significantly enhanced in the responders to the IL-23 inhibitor, as compared with that in non-responders. Taurine, an abundant amino acid in leukocytes, is found in high concentrations in inflammatory lesions and tissues exposed to oxidative stress. [53]. Collectively, these findings and our data imply that a shift in gut bacterial composition due to the IL-23 inhibitor could lead to significant changes in taurine metabolism, which may correlate with an improvement in the inflammatory status in patients with psoriasis.

Our results should be considered in the context of several limitations. First, sample sizes were limited, and larger cohorts should be assessed in future studies. Second, due to the relatively limited resolution of the 16S rRNA sequencing technique [54], shotgun metagenomic sequencing methods are needed to identify specific bacterial strains in psoriasis. Third, based on the gut-microbiota-mediated metabolic pathways related to the response to the IL-23 inhibitor or IL-17 inhibitors identified in psoriatic patients, it is necessary to explore their key regulatory targets. Finally, we did not investigate inflammatory markers collected from the peripheral blood, gut, and stool so we could not explain the association of inflammatory changes with the microbial composition.

In summary, treatments with IL-23 and IL-17 inhibitors were associated with distinct shifts in gut microbial composition in patients with psoriasis. Significant differences in the relative abundance of bacteria taxa between the responders and non-responders suggested that IL-23 and IL-17 inhibitors may functionally interact with gut microbiota to reduce cutaneous inflammation. Moreover, we demonstrated the association between the treatment response and gut microbial function, which might serve as potential biomarkers in the treatment response.

## 4. Material and Methods

### 4.1. Study Design and Patients

This prospective study enrolled forty-eight patients with psoriasis, including 30 cases treated with an IL-23 inhibitor (guselkumab) and 18 cases with IL-17 inhibitors (ixekizumab or secukinumab) in the Chang Gung Memorial Hospital (Taoyuan, Taiwan) from September 2020 to March 2022. None of the included cases had taken systemic antibiotics, systemic immunosuppressant agents, oral corticosteroids, and probiotics one month before each sample collection other than guselkumab, ixekizumab, or secukinumab. The anti-IL-23 medication group received guselkumab 100 mg at week 0, 4, and every 8 weeks thereafter. The anti-IL-17 medication group received either ixekizumab 160 mg at week 0, 80 mg at week 2, 4, 6, 8, 10, and 12, and 80 mg every 4 weeks thereafter, or secukinumab 300 mg at week 0, 1, 2, 3, and 4, and every 4 weeks thereafter. The demographics and clinical data of the patients, including their age, gender, weight, and psoriatic arthritis (PsA), were collected at baseline. Psoriasis Area and Severity Index (PASI) score and serum C-reactive protein (CRP) level were collected at week 0, 4, 12, and 24. Responders were defined as those having a PASI improvement of ≥90% after 24 weeks of treatment and non-responders as having a PASI improvement of <90%. Information about food intake during the study was collected at week 0, 12, and 24 through a food-frequency questionnaire (FFQ) [55].

### 4.2. DNA Isolation and 16S rRNA Gene Sequencing

Stool specimens were collected using the Longsee Fecalpro Kit (Longsee Medical Technology Co., Guangzhou, China) at baseline and 4 weeks, 12 weeks, and 24 weeks after treatment. As described previously [56], DNA was isolated by using the QIAamp PowerFecal Pro DNA Kit for Feces (Qiagen, Germantown, MD, USA) following the manufacturer’s instructions. Around 0.25 g of the sample in the Bead Tube was added with 750 μL of PowerBead Solution and 60 μL of Solution C1, which was then heated at 65 °C for 10 min. The mixture was vortexed by using a PowerLyser Homogenizer at 1000 RPM for 10 min. After the steps of cell lysis, removal of contaminating matters, washing and eluting with DNA-free, PCR-grade water, DNA was extracted. The concentrations and qualities of the extracted DNA were measured by using Qubit 4 Fluorometer (Thermo Fisher Scientific, Waltham, MA, USA).

The variable regions 3 and 4 (V3–V4) of 16S rRNA gene were PCR (polymerase chain reaction)-amplified by using the primer set (the Illumina V3 forward 5′-CCTACGGGNGGCWGCAG-3′ and V4 reverse 5′-GACTACHVGGGTATCTAATCC-3′) [57]. The Illumina sequencing adapters ligated to the purified amplicons by a second-stage PCR using the TruSeq DNA LT Sample Preparation Kit (Illumina, San Diego, CA, USA) were performed to construct a library. Purified libraries were quantified, normalized, pooled, and applied for cluster generation and sequencing on a MiSeq instrument (Illumina).

### 4.3. Sequencing Data Processing and Species Annotation

Paired-end reads were processed by using DADA2 [58] to filter out noisy sequences, correct errors in marginal sequences, remove chimeric sequences, and eliminate singletons to infer amplicon sequence variants (ASVs). Bacterial taxonomy was assigned by applying a pre-fitted QIIME2 classifier built with the Scikit-lean package [59] based on the information collected from the SILVA database [60]. Arrangement of multiple sequences were performed by the PyNAST software v.1.2 [61] for assessment of the phylogenetic relationship of various ASVs, and a phylogenetic tree was constructed with the FastTree 2.1.0 [62].

### 4.4. Microbial Gene Function Prediction

Functional composition of metagenomes was predicted from 16S rRNA data by the Tax4Fun2 software [63]. To predict functional profile of the microbial community, the taxonomic abundance transformed from the SILVA-based 16S rRNA and normalized by the 16S rRNA copy number acquired from the NCBI annotations were applied to incorporate the precomputed functional profiles of KEGG pathways [63]. KEGG analysis was only focused on “Metabolism” pathways.

### 4.5. Statistical Analysis

Demographic and clinical characteristics were presented as *n* (%) for categorical variables and mean  ±  standard deviation (SD) or median with range for continuous variables. For estimating alpha diversity, species richness was evaluated by inverse Simpson’s index. Beta diversity was analyzed by Bray–Curtis or unweighted-UniFrac distance matrix. In order to investigate the association of treatment effect and bacteria in the fecal specimens, we further identified differentially abundant bacterial taxa among groups. Statistically significant biomarkers were analysed by the non-parametric Kruskal–Wallis test, Wilcoxon rank-sum test, and linear discriminant analysis (LDA) to identify differentially abundant taxa. Change in relative abundance after treatment from the baseline was compared between responders and non-responders by fitting a linear mixed model, measured on a continuous scale to identify longitudinal biomarkers. All statistical tests are two-tailed, and a *p*-value less than 0.05 was considered statistically significant.

## Figures and Tables

**Figure 1 ijms-24-04568-f001:**
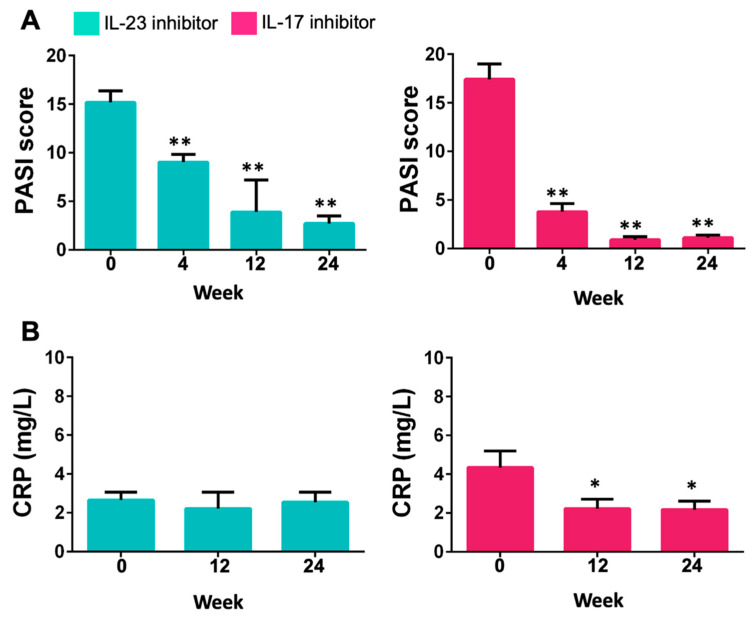
PASI scores (**A**) and CRP levels (**B**) at different time points after the treatment with IL-23 (guselkumab) or IL-17 (secukinumab and ixekizumab) inhibitor. * *p* < 0.05 versus week 0 for each group; ** *p* < 0.001 versus week 0 for each group.

**Figure 2 ijms-24-04568-f002:**
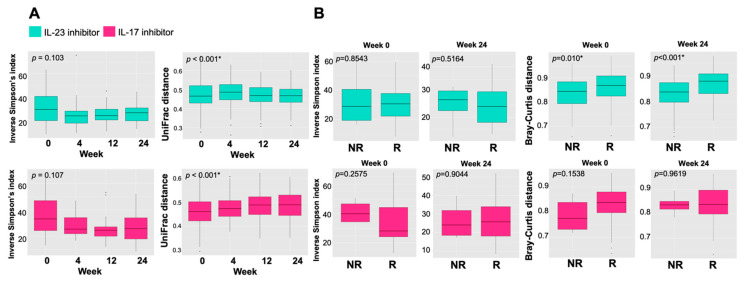
Gut microbial composition in psoriatic patients during the 24 weeks of treatment with IL-23 (guselkumab) or IL-17 (secukinumab and ixekizumab) inhibitors. (**A**) The distribution of α-diversity (Inverse Simpson’s index) and β-diversity (UniFrac distance) of gut microbiota at weeks 0, 4, 12, and 24 in patients treated with IL-23 or IL-17 inhibitors. (**B**) The distribution of α-diversity (Inverse Simpson’s index) and β-diversity (Bray–Curtis distance) at weeks 0 and 24 between responders (R) and non-responders (NR) of the two groups. * *p* < 0.05.

**Figure 3 ijms-24-04568-f003:**
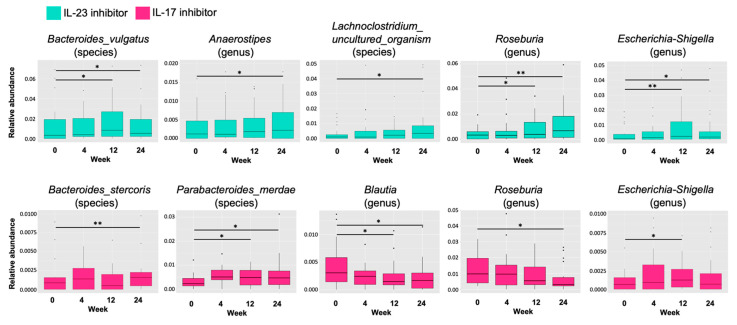
The relative abundance of specific gut microbial taxa during 24 weeks of treatment with IL-23 (guselkumab) or IL-17 (secukinumab and ixekizumab) inhibitors. Data are shown at each time point as box plots and represent 25–75% interquartile range of the median. (* *p* < 0.05, ** *p* < 0.005).

**Figure 4 ijms-24-04568-f004:**
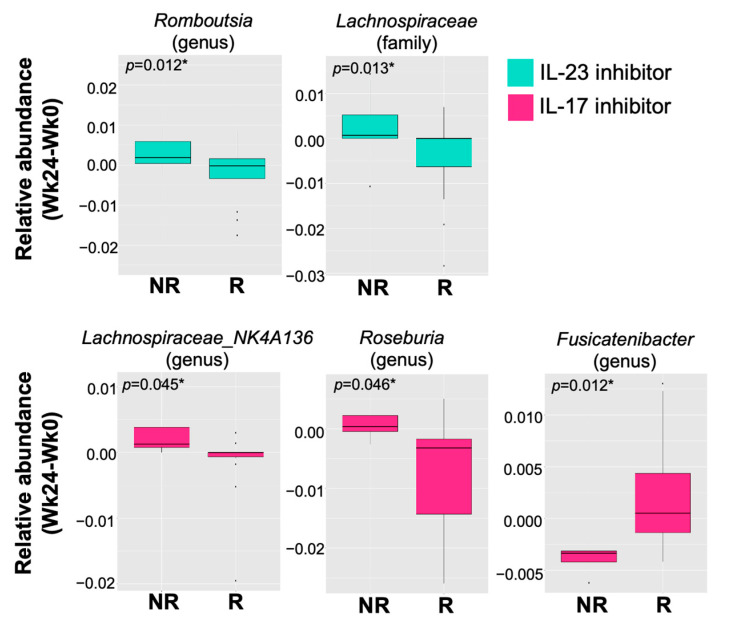
Change in relative abundance of specific gut microbial taxa from the baseline between responders (R) and non-responders (NR) treated with IL-23 (guselkumab) or IL-17 (secukinumab and ixekizumab) inhibitors. The altered level of gut microbiomes at family, genus, or species level at week 24 from the baseline was significantly different between responders and non-responders treated with IL-23 or IL-17 inhibitors. * *p* < 0.05.

**Table 1 ijms-24-04568-t001:** Demographics and characteristics of study groups.

Variables	All (*n* = 48)	IL-23 Inhibitor (Guselkumab) (*n* = 30)	IL-17 Inhibitor (Secukinumab and Ixekizumab (*n* = 18)	*p*-Value ^a^
Ages (years), mean ± SD	48.1 ± 11.9	45.2 ± 11.6	52.8 ± 12.6	0.045 *
Gender/male, *n* (%)	41 (85.4%)	25 (83.3%)	16 (88.9%)	0.696
Weight (kg), mean ± SD	75.2 ± 11.6	74.7 ± 12.5	76.1 ± 10.2	0.675
Psoriatic arthritis, *n* (%)	15 (31.3%)	8 (26.7%)	7 (38.9%)	0.522
PASI score, mean ± SD	16.0 ± 6.6	15.2 ± 6.5	17.4 ± 6.7	0.273
PASI-90 at wk24, *n* (%)	31 (64.6%)	17 (57.0%)	14 (77.0%)	0.214
CRP (mg/L), mean ± SD	3.3 ± 2.8	2.7 ± 2.3	4.4 ± 3.6	0.084

IL, interleukin; PsA, psoriatic arthritis; PASI, Psoriasis Area Severity Index; CRP, C-reactive protein. ^a^ The independent sample *t*-test or non-parametric Wilcoxon rank-sum test were used for intergroup comparisons. * *p* < 0.05.

**Table 2 ijms-24-04568-t002:** Comparison of KEGG metabolism pathways of fecal microbiota between week 0 and week 24 post-treatment in psoriasis.

Treatment	KEGG Pathway	Week 0	Week 24	*p*-Value
IL-23 inhibitor	Lipid metabolism			
	Fatty acid biosynthesis [PATH:ko00061]	0.00864	0.008873	0.002367
	00071 Fatty acid degradation [PATH:ko00071]	0.004989	0.005184	0.04726
	00565 Ether lipid metabolism [PATH:ko00565]	0.000109	0.000132	0.03842
	00592 alpha-linolenic acid metabolism [PATH:ko00592]	5.31 × 10^−5^	6.27 × 10^−5^	0.04265
	Energy metabolism			
	00195 Photosynthesis [PATH:ko00195]	0.005238	0.005149	0.02341
	00680 Methane metabolism [PATH:ko00680]	0.01616	0.01592	0.007612
	00710 Carbon fixation in photosynthetic organisms [PATH:ko00710]	0.01162	0.01144	0.002367
	Amino acid metabolism			
	00220 Arginine biosynthesis [PATH:ko00220]	0.01214	0.01188	0.01205
	00270 Cysteine and methionine metabolism [PATH:ko00270]	0.02268	0.0225	0.04049
	00480 Glutathione metabolism [PATH:ko00480]	0.003758	0.003977	0.0221
	Carbohydrate metabolism			
	00051 Fructose and mannose metabolism [PATH:ko00051]	0.01362	0.01336	0.0106
	00562 Inositol phosphate metabolism [PATH:ko00562]	0.001421	0.001496	0.03272
	00332 Carbapenem biosynthesis [PATH:ko00332]	0.001633	0.001594	0.04971
IL-17 inhibitor	00300 Lysine biosynthesis [PATH:ko00300]	0.01177	0.01159	0.03423
	00901 Indole alkaloid biosynthesis [PATH:ko00901]	1.91 × 10^−6^	2.50 × 10^−6^	0.03036

**Table 3 ijms-24-04568-t003:** Comparison of difference in abundance of KEGG metabolism pathways from the baseline to week 24 post-treatment between responders and non-responders.

Treatment	KEGG Pathway	Responder	Non-Responder	*p*-Value
IL-23 inhibitor	Taurine and hypotaurine metabolism [PATH:ko00430]	0.0001581	−7.16 × 10^−5^	0.04808
IL-17 inhibitor	Amino acid metabolism			
	Glycine, serine, and threonine metabolism [PATH:ko00260]	0.0002248	−0.0007905	0.02484
	Valine, leucine, and isoleucine biosynthesis [PATH:ko00290]	9.18 × 10^−5^	−0.0009196	0.04641
	Phenylalanine, tyrosine, and tryptophan biosynthesis [PATH:ko00400]	3.27 × 10^−5^	−0.001027	0.04641
	Cysteine and methionine metabolism [PATH:ko00270]	−7.21 × 10^−5^	−0.0007994	0.03464
	Lysine biosynthesis [PATH:ko00300]	−7.21 × 10^−5^	−0.0007994	0.03464
	Biosynthesis of other secondary metabolite			
	Novobiocin biosynthesis [PATH:ko00401]	3.53 × 10^−5^	−0.0001187	0.03464
	Acarbose and validamycin biosynthesis [PATH:ko00525]	8.18 × 10^−5^	−0.0001209	0.01765
	Carbohydrate metabolism			
	Citrate cycle (TCA cycle) [PATH:ko00020]	0.0004246	−0.001106	0.04641
	C5-Branched dibasic acid metabolism [PATH:ko00660]	9.64 × 10^−5^	−0.000547	0.04641
	One carbon pool by folate [PATH:ko00670]	0.0001271	−0.000506	0.01765
	Glucosinolate biosynthesis [PATH:ko00966]	1.24 × 10^−5^	−0.0001168	0.02484
	Synthesis and degradation of ketone bodies [PATH:ko00072]	−4.68 × 10^−6^	0.0002448	0.02484
	Biosynthesis of vancomycin group antibiotics [PATH:ko01055]	5.35 × 10^−5^	−9.27 × 10^−5^	0.02484
	Sphingolipid metabolism [PATH:ko00600]	0.0003941	−0.0008993	0.03464
	Glycosphingolipid biosynthesis—globo and isoglobo series [PATH:ko00603]	0.0002465	−0.000696	0.03464
	Aminobenzoate degradation [PATH:ko00627]	3.99 × 10^−5^	0.0002052	0.04641

## Data Availability

The data presented in this study are available on request from the corresponding author.

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
