# Peer review of "Compositional Alteration of Gut Microbiota in Psoriasis Treated with IL-23 and IL-17 Inhibitors"

_ijms, 2023, doi:10.3390/ijms24054568_

Round 1
Reviewer 1 Report
In this article the authors present a study investigating the effect of guselkumab and IL-17 inhibitor treatment on the gut microbiota over a time period of 24 weeks. They observed changes in the β-diversity over time and alterations in gut microbiota composition that were at least partly not shared between the treatment groups.
Overall, the study design is clear and of relevance to the scientific community, and the data presentation is good and the manuscript is written well.
However, the study would profit from some adaptations in order to enhance the relevance of this work.
Major points:
a) Title: the term functional should be removed as there were no functional test performed.
b) I miss information of clinical patient characteristics (e.g. disease duratiuon) and there effect on the microbiota at baseline as these could potentially bias the analysis therafter.
c) Investigating inflammatory markers from the gut or stool would be very informative as this could link the changes in microbiota composition to inflammatory modes of action.
d) Along the same line: An analysis of peripheral blood (e.g. immun cell composition) or peripheral inflammatory markers (e.g. cytokines) would be helpful in order to evaluate the importance oft he gut microbial changes.
Minor points:
e) In the Introduction the appearance of refs 11-13 should be reformatted.
f) In the text fig. 3A and 3B are mentioned, but in the figure itself there is no a or b.
Author Response
Please see the attchment.

Reviewer 2 Report
The authors studied the gut microbiota of patients treated with biologics. The work is somehow interesting, however, I have several concerns:
1. The title and next after: The authors of the paper should either discuss individual molecules separately or refer to the entire group of drugs. Thus, they should compare anti-IL23 vs. anti-IL17 drugs or guselkumab vs. ixekizumab vs. secukinumab.
2. The group of patients treated with IL-17 blockers is small and the results may be thus biased by many factors. Please provide information of any other treatment taken by the patients, the diet, as well as whether they changed their usual habits during the treatment.
3. Although statically significant changes were observed in guselkumab group during the treatment, there is a significant overlap of the achieved results (figure 2) - therefore the authors should be very careful while interpreting their results and drawing conclusions.
Please correct the word: secukinjmab
Round 2
Reviewer 1 Report
Unfortunately, the authors were not able to add new data in regard to inflammation to the current work. Anyway, I feel the adaptations incorporated in the revised version of the manuscript improved the overall quality. Therefore, I feel that the work is now suitable for publication.
Reviewer 2 Report
Dear Editor,
The manuscript has been sufficiently improved.
Kind regards,
Adam Reich